# Redesigning Primary Care: The Emergence of Artificial-Intelligence-Driven Symptom Diagnostic Tools

**DOI:** 10.3390/jpm13091379

**Published:** 2023-09-15

**Authors:** Christian J. Wiedermann, Angelika Mahlknecht, Giuliano Piccoliori, Adolf Engl

**Affiliations:** 1Institute of General Practice and Public Health, Claudiana—College of Health Professions, 39100 Bolzano, Italy; 2Department of Public Health, Medical Decision Making and HTA, University of Health Sciences, Medical Informatics and Technology-Tyrol, 6060 Hall, Austria

**Keywords:** primary health care, artificial intelligence, symptom assessment, telemedicine, patient satisfaction

## Abstract

Modern healthcare is facing a juxtaposition of increasing patient demands owing to an aging population and a decreasing general practitioner workforce, leading to strained access to primary care. The coronavirus disease 2019 pandemic has emphasized the potential for alternative consultation methods, highlighting opportunities to minimize unnecessary care. This article discusses the role of artificial-intelligence-driven symptom checkers, particularly their efficiency, utility, and challenges in primary care. Based on a study conducted in Italian general practices, insights from both physicians and patients were gathered regarding this emergent technology, highlighting differences in perceived utility, user satisfaction, and potential challenges. While symptom checkers are seen as potential tools for addressing healthcare challenges, concerns regarding their accuracy and the potential for misdiagnosis persist. Patients generally viewed them positively, valuing their ease of use and the empowerment they provide in managing health. However, some general practitioners perceive these tools as challenges to their expertise. This article proposes that artificial-intelligence-based symptom checkers can optimize medical-history taking for the benefit of both general practitioners and patients, with potential enhancements in complex diagnostic tasks rather than routine diagnoses. It underscores the importance of carefully integrating digital innovations while preserving the essential human touch in healthcare. Symptom checkers offer promising solutions; ensuring their accuracy, reliability, and effective integration into primary care requires rigorous research, clinical guidance, and an understanding of varied user perceptions. Collaboration among technologists, clinicians, and patients is paramount for the successful evolution of digital tools in healthcare.

## 1. Introduction

Modern health care is critical. The convergence of burgeoning patient demands, primarily due to a progressively aging population and a diminishing general practitioner (GP) workforce, has rendered access to primary care increasingly challenging [1,2]. This situation is exacerbated by the prevalence of non-urgent medical consultations which unnecessarily strain the system. Consequently, we face a paradoxical situation in which an escalation in technological advancements is countered by decreasing patient satisfaction [3,4].

The COVID-19 pandemic has provided unexpected observations, highlighting the efficacy of alternative consultation methods. The pandemic-induced decline in face-to-face visits highlights the prospect that some treatments, which are now deemed superfluous, potentially carry the risk of iatrogenic harm. Such revelations underscored the opportunity to minimize unnecessary care, thereby safeguarding patient well-being and fortifying the sustainability of healthcare [5].

This article discusses the role and potential of AI-driven symptom checkers [6], particularly in light of the challenges confronting modern healthcare. We aimed to evaluate the efficiency, utility, and challenges of integrating these digital tools into primary care settings by drawing insights from a recent study conducted in Italian general practices [7]. The perspectives of both physicians and patients regarding this emergent technology were elucidated, shedding light on its potential advantages and pitfalls.

## 2. Symptom Checkers

Symptom checkers with chatbots are digital tools that use AI algorithms to engage users in a conversational interface that allows them to input and describe their medical symptoms. These tools then analyze the provided information to offer potential differential diagnoses, provide triage recommendations, and suggest appropriate next steps in care [8]. Designed to be user-friendly and accessible, they offer patients an initial point of contact for health concerns, helping to guide them towards appropriate medical care or self-management options [9].

In the evolving healthcare landscape, the role of AI is slowly gaining recognition [10]. AI-driven symptom checkers based on chatbots are being explored as potential tools for addressing ongoing challenges in the sector. These digital platforms, which merge algorithmic analyses with user interfaces, are seen as possible aids in complementing the work of GPs. Their tentative benefit may be to offer patients a preliminary platform for self-assessment, possibly assisting in their healthcare decisions. Such tools might offer GPs an additional layer of information, potentially allowing them to allocate more time to intricate cases [9]. However, recent studies have shed light on physicians’ perceptions regarding the extensive use of AI in primary care [11]. A significant proportion of GPs perceive the potential of AI as somewhat constrained, a sentiment that contrasts with the optimism of biomedical informaticians. Furthermore, the sophisticated framework underlying symptom checkers has added to the allure of these digital tools. Not only are they designed to collect preliminary data, but they also possess the capability to provide differential diagnoses based on inputted symptoms. This feature becomes pivotal in guiding patients towards appropriate medical action, whether the tool recommends immediate medical attention or considers alternative treatment pathways [12]. Essentially, these tools may play a crucial role in the triage process, helping prioritize cases based on urgency and clinical relevance. Such functionalities could dramatically reshape the way primary care operates, optimize resource allocation, and ensure timely medical intervention.

However, each revolutionary tool presents challenges that cannot be ignored. Symptom checkers suggest potential causes and recommend courses of action based on symptoms. However, if their output does not align with the users’ personal experiences or falls short of their expectations, it might lead to unwarranted healthcare-seeking behaviors [13,14]. The effectiveness and reliability of symptom checkers are subject to intense scrutiny [15,16]. Although they undoubtedly present a revolutionary approach to preliminary medical assessments, the crux of their value lies in their ability to provide accurate and safe advice.

Previous research has shed light on areas of concern. Causal reasoning remains a vital missing component for applying machine learning to medical diagnoses [17]. Some studies have highlighted the propensity of these tools to either misdiagnose or lean towards excessively cautious triage recommendations [9,16].

The diagnostic and triage capabilities of symptom checkers remain limited, especially in non-urgent primary care situations [9,18]. A study revealed that most laypersons performed better than symptom checkers when assessed using clinical vignettes, although the symptom checkers were more reliable in identifying emergency cases [19]. These potential pitfalls serve as cautionary notes, emphasizing the need for rigorous validation, continuous updates, and user education to ensure that symptom checkers realize their full potential without compromising patient safety.

### Insights from the Italian Experience

Amid rapid advancements in AI, the Italian healthcare system has actively explored the integration of AI-driven symptom checkers into primary care. The study in question [7], conducted in northern Italian GP offices, particularly during the challenging times of the pandemic, is among the first of its kind. Specifically designed as a feasibility study, the research focused on ten general practitioners (GPs) and the patients visiting their offices.

The patients were prompted to use a chatbot-based symptom checker before their medical visits. This checker not only facilitated anamnestic screening for COVID-19 but also employed a medical history algorithm tailored to the patient’s specific medical problem. The data entered were then relayed to the GP, serving as an auxiliary medical history aid. After their medical consultations, both the participating physicians and their patients were tasked with evaluating the symptom checker based on their experience. Of the 225 patients who participated in the study, 145 completed the post-visit survey; however, after excluding 29 patients due to a chatbot-estimated medium or high anamnestic risk for COVID-19, a total of 116 post-visit questionnaires were included in the final analysis. The patients were predominantly female (55%), with a median age of 47 years. Most had vocational schooling (39%) or were high school graduates (28%). A vast majority (87%) relied on their GP for health information, while 14% used online sources. Health assessments varied, with 44% considering their health ‘very good’ and 15% marking it ‘average’. To ensure a comprehensive evaluation, the physicians also offered a final overarching review of the symptom checker upon completing the practice phase. Table 1 presents the main findings.

From the results, it became evident that while most patients had not previously engaged with symptom checkers, those who had regarded them positively. Specifically, almost half of the patients and a quarter of the doctors said they were ‘fairly’ or ‘very’ satisfied. When asked to provide reasons for their opinions, the patients applauded the checker’s ease of use, precise questioning, time-saving potential, and the tool’s capacity to encourage self-reflection. Interestingly, every second patient expressed a willingness to use the symptom checker at home, viewing it as a potential means of assessing initial health concerns, minimizing unnecessary medical visits, and assisting their physicians. Demographics such as age, sex, and education level did not play a significant role in shaping patients’ attitudes towards the symptom checker. Notably, the vast majority of participants believed that the tool did not affect the duration of the medical visit. Only a marginal fraction felt that the tool might disrupt the quality or flow of the consultation [7].

An Italian study provided insights into the reception of symptom checkers, revealing varied perspectives between patients and doctors. While there is interest in such tools, it is still uncertain whether they will become standard in primary care. This study emphasizes that careful clinical guidance is crucial before considering wider adoption.

## 3. Discussion

Recent study results [9,18,20], further illuminated by an Italian publication [7], underscore the importance of meticulous clinical guidance in the evolution of symptom checkers. Symptom checkers have emerged as a potential avenue for addressing challenges in primary care, such as the growing workload due to an aging population and the declining number of GPs. Their adoption could increase healthcare efficiency and possibly relieve GPs; however, their broad implementation demands a robust, evidence-based evaluation of their efficacy, safety, and cost-effectiveness.

An Italian study [7] pointed out a dichotomy in attitudes towards symptom checkers: patients tend to view them positively, finding empowerment in controlling their health, whereas some GPs see them as rather unhelpful in relation to the patients’ self-management or reducing unnecessary visits.. This difference in perception stresses the importance of understanding both patients’ and GPs’ experiences with digital tools. Trust in symptom checkers might differ between patients and GPs based on their respective vulnerabilities, the anticipation of the AI’s decisions, an understanding of the ‘contract’ with the AI, an evaluation of the AI’s trustworthiness, and what is needed from explainable AI [21]. Patients might have more surface-level interactions, trusting the system to provide guidance, whereas GPs, with their deeper medical knowledge, might scrutinize and evaluate the AI’s suggestions more critically. Interestingly, neither patients nor GPs perceived the use of symptom checkers as significantly time-averse during medical consultations. Moreover, the varied levels of satisfaction among GPs in the study, especially in relation to post-visit evaluations and the concluding survey, hint at external factors that influence GPs’ opinions [7]. From the patient’s standpoint, the use of symptom checkers adds an element of novelty without incurring significant effort. On the other hand, for GPs, especially during high-workload periods such as the COVID-19 pandemic, integrating these tools with their tasks potentially skewing their satisfaction rates.

However, one constant was persistent skepticism regarding the diagnostic accuracy of symptom checkers. The divergence in perceptions of the value of symptom checkers between GPs and patients hints at broader challenges in healthcare, particularly regarding the emphasis on human touch and the tactile aspects of the diagnostic process. While digital advancements in patient-GP communication are highly sought after, with patients voicing a strong desire for features such as direct messaging platforms, streamlined appointment scheduling, and efficient symptom tracking, it is imperative to note the irreplaceable value that patients place on face-to-face consultations and the significance of maintaining eye contact during visits [22]. The lack of consistent evidence regarding the workload-reducing potential of symptom checkers requires further research. Balancing digital innovations with human touch is important for optimizing patient care. The highlighted interest of about half of the patients in the chatbot’s use for pre-visit preparations suggests a potential avenue for enhancing patient-GP interactions. The chatbot’s positive effect on the patients’ self-reflection and attentiveness during medical consultations is worth noting.

While patients appreciated the symptom checkers’ speed and user-friendly interface in an Italian study, GPs reported observing technical and procedural challenges among patients. This discrepancy underscores the need to design tools that are both intuitive and user-centric. In addition, the data hinted at a possible age bias in the acceptance and usage of symptom checkers, suggesting that a more tailored approach for diverse populations might be beneficial. In light of the observed differences between the patients’ and GPs’ attitudes towards AI-based symptom checkers, strategies could be pursued to bridge this divide. Firstly, it is pertinent to emphasize the importance of iterative feedback. By establishing mechanisms that enable GPs to consistently convey their experiences and hurdles with these tools to AI developers, we can ensure that these symptom checkers are fine-tuned in real time to better serve clinical needs. On the patient front, refining the user interfaces of AI tools to be more intuitive can heighten their appeal and usability. A design centered around the patient experience ensures that the tools are more aligned with their expectations and comfort levels. Lastly, a clearer insight into the algorithms underpinning AI tools can dispel reservations and foster deeper trust. By elucidating how these systems arrive at specific conclusions, we can instill greater confidence in both patients and doctors. Implementing these strategies could prove instrumental in harmonizing perceptions and facilitating a smoother integration of AI-driven symptom checkers into primary care.

Our findings are consistent with a growing body of literature that examines the utility, efficacy, and challenges of using AI and decision support systems (DSSs) in primary care settings. Gottliebsen and Petersson [15] investigated the use of intelligent online triage tools in primary care and observed that the current systems might be underdeveloped, providing limited benefits. This resonates with our observations of the challenges of seamlessly integrating such tools into routine clinical practice. Moreover, Semigran et al. [9] conducted an audit on the diagnostic and triage accuracy of online symptom checkers. They found that these systems often lacked diagnostic accuracy albeit being risk-averse, thus potentially directing patients to seek unnecessary medical attention. Such findings echo concerns about overburdening healthcare infrastructures with avoidable patient visits. In addition, a systematic review [16] discussed the uncertainties surrounding digital symptom checkers and their potential impact on healthcare outcomes. These authors noted the specific preference of younger and more educated populations for online and digital services, emphasizing the implications for health equity.

There are studies examining the broader implications and potentials of differential diagnosis DSSs in primary care. McParland et al. [23] indicated potential roles for such DSS in assisting both clinicians and the public; however, the design and implementation considerations must cater to the specific needs of these groups. Kostopoulou et al. [24] highlighted the opportunity for DSSs to combat the incompleteness and biases prevalent in routine primary care data. Such findings underscore the importance of a holistic approach when designing and implementing these systems. Furthermore, while the utility of some online diagnostic systems like Isabel in general practice has been assessed, some have suggested the need for further modifications to ensure their suitability in primary care contexts [25]. This mirrors our study’s emphasis on the customization and adaptability of AI tools for their intended clinical settings.

In conclusion, while there is interest and potential in the domain of AI and DSS in primary care, there are evident challenges and considerations. Our study, in light of others like those cited above, reinforces the importance of thorough evaluations, iterative design processes, and stakeholder engagement in this rapidly evolving landscape.

This opinion article offers an exploration of AI-driven symptom checkers in primary care, drawing insights primarily from a recent study [7]. Though it provides valuable perspectives, its limitations include the subjective nature of the content, potential biases from the focus on a specific demographic, and concerns about the generalizability of the findings, particularly given the rapidly evolving nature of AI. The highlighted value of face-to-face consultations and the diagnostic accuracy of symptom checkers require further exploration. Moreover, the unique circumstances of the COVID-19 pandemic may have influenced the observations, questioning their post-pandemic relevance. A broader empirical foundation and consideration of various settings can enhance the insights presented here.

A pivotal question surrounding symptom-checkers with chatbots revolves around their capacity to overcome present limitations. Specifically, can these tools, through continued development, successfully replicate the causal reasoning of a human expert? As AI evolves, there is a growing optimism that future iterations of AI systems will improve in emulating complex human reasoning processes. However, it remains uncertain whether technology can ever truly capture the nuanced and multifaceted nature of human cognitive abilities. Thus, while advancements are anticipated, the extent to which these tools can match human intuition and reasoning is still a topic of debate.

## 4. Conclusions

AI-driven symptom checkers have emerged as promising tools for addressing primary care challenges. Recent findings, especially from an Italian context, demonstrate a dichotomy in perspectives: while patients appreciate the empowerment and user-friendliness of these tools, some GPs voice concerns, particularly regarding the tools’ diagnostic accuracy. Neither group perceived significant time-saving benefits during consultations. Emphasizing human touch remains paramount despite the push for digital innovations in patient-GP communication. Further, the highlighted interest of the patients in using chatbots for pre-visit preparations hints at enhancing patient-GP interactions.

The potential of AI-based symptom checkers becomes evident, especially when considering the optimization of medical-history taking. For GPs, these tools can streamline the process, making it more efficient and focused. For patients, they can be empowering, offering a sense of agency and participation. However, it is recognized that in routine general practice, the challenge is not so much diagnostic difficulty for common ailments but rather the increasing demands of rare diseases. Given this issue, while AI tools can aid in routine diagnostics, their real potential lies in assisting with the more complex and intricate diagnostic tasks. By concentrating on these challenging areas, AI can significantly complement the expertise of GPs, leading to more accurate and timely interventions.

The study underscores the need for symptom checkers to be intuitive and user-centric and the importance of rigorous validation to ensure patient safety. These observations warrant further exploration. Future research should aim for a broader empirical foundation across various settings to fully capture the potential challenges of AI in primary care.

## Figures and Tables

**Table 1 jpm-13-01379-t001:** Comparative perspectives on symptom checkers from patients and GPs in Italian primary care [7].

Variable	Patients’ Perspectives	GPs’ Perspectives
Experience with Symptom Checkers	Most had not previously used a symptom checker Positive feedback on the ease of use	Varied experiences and perspectives
Satisfaction	49% were ‘rather’ or ‘very’ satisfied	27% were ‘rather’ or ‘very’ satisfied
Usefulness	Precise questioning Time-saving potential Encourages self-reflection	Value as an auxiliary medical history aid
Willingness to Use	50% are willing to use it at home	Concerns about additional workload
Impact on Medical Visit Duration	75% felt no impact	84% felt no impact
Trust in AI	Surface-level interaction General guidance	AI suggestions should be scrutinized and evaluated more critically

Abbreviations: AI, artificial intelligence; GPs, general practitioners.

## Data Availability

No new data were created.

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
