# Peer review of "Redesigning Primary Care: The Emergence of Artificial-Intelligence-Driven Symptom Diagnostic Tools"

_jpm, 2023, doi:10.3390/jpm13091379_

Round 1
Reviewer 1 Report
1. The contributions of this study should be stated.
2. How many patients received feedback?
3. Was a survey administered to the patients?
4. If a survey was conducted, why were these survey samples not presented?
5. If more results on demographic information are obtained, they should be added.
Reviewer 2 Report
This opinion paper discussed the function of AI in primary care. It mainly focused on the published paper, which investigated the attitudes of patients and general practitioners to AI-based symptom checkers. I have the following concerns.
1. The full name of AI should be provided in the text.
2. As an opinion paper, authors’ opinion is not clear. For or disagree AI-based symptom checkers. If agree, please list the merits and disadvantages and further give brief opinions to overcome these disadvantages.
3. The focused paper (Ref. 7) stated different attitudes of patients and general practitioners. This paper gave the reasons why the different attitudes were generated. Could authors give some opinions for eliminate differences.
4. I do not know whether other studies on the similar topics of Ref. 7 exist. If exist, could authors employed them to strengthen the current manuscript.
Round 2
Reviewer 1 Report
The publication of the article is appropriate.
Reviewer 2 Report
All my comments have been addressed. The current version can be accepted.